# Survey of clot waveform analysis of normal activated partial thromboplastin time in patients with cirrhosis and sepsis at Le Van Thinh hospital

Tran Qui Phuong Linh[1], Nguyen Thi Nhan![ORCID][2]*, Le Minh Thuan[3], Le Trung Phuong[4]

**1** Department of Hematology - Blood Transfusion, Le Van Thinh Hospital, Thu Duc city, Ho Chi Minh city, Vietnam, **2** Medical Laboratory Faculty, Nguyen Tat Thanh University, Ho Chi Minh city, Vietnam, **3** Ho Chi Minh City Medicine and Pharmacy University, Ho Chi Minh city, Vietnam, **4** Institute of Technology and Development ITD, Ha Noi city, Vietnam

* ntnhan@ntt.edu.vn

## Abstract

This study aimed to establish the correlation between activated partial thromboplastin time (aPTT) in seconds and ratio, first derivative peak (Max1), second derivative peak (Max2), second derivative minimum peak (Min2), and delta-OD of clot waveform analysis in patients diagnosed with cirrhosis, sepsis, and a control group. The study involved a series of cases using the Bayesian ANOVA model with Bayes factor (BF) estimated using the Markov Chain Monte Carlo sampling method with 2 chains. Each chain consisted of 10 000 repetitions and started 5 000 times to predict the variables aPTT (s), aPTT (ratio), Max1, Max2, Min2, and delta-OD respectively across the 3 study groups. The model used was variable ~ stage – 1. This analysis illustrates the adjustment of the posterior proportion for multiple testing by setting the prior probability of the initial hypothesis to 0.5 for all comparisons. There is insufficient evidence that aPTT (s), aPTT (ratio), and Min2 differ between the study groups (0 < BF < 3). There is compelling evidence for the Max1 value in both the sepsis group and the control group (BF10 = 44.042). Additionally, between the sepsis group and the cirrhosis group, there is substantial evidence for the Max1 value (BF10 = 5.842). Moreover, there is convincing evidence for the Max2 value in the sepsis group compared to the control group (BF10 = 38.782). There is almost confirmatory evidence that the delta-OD value in the sepsis group is higher than that in the control group (BF10 = 1251.781). Finally, there is convincing evidence that the sepsis group has a higher delta-OD value than the cirrhosis group (BF10 = 35.555). The Max1, Max2, and delta-OD indices effectively reflect coagulation disorders in patients with cirrhosis and sepsis, even when the aPTT test results are within normal limits.

**Data availability statement:** All relevant data are within the manuscript and its Supporting Information files.

**Funding:** The author(s) received no specific funding for this work.

**Competing interests:** The authors have declared that no competing interests exist.

## Introduction

Sepsis is a life-threatening condition caused by an imbalance in the body's response to infection, leading to organ dysfunction. Organ dysfunction in sepsis can affect the circulatory, renal, hepatic, respiratory, and coagulation systems [1]. The immune and coagulation systems also interact with each other. Research suggests that inflammation triggers coagulation activation, and vice versa, coagulation significantly influences inflammatory activity [2]. During the injured or infected phase, the exposed tissue factors come in contact with blood, releasing pro-inflammatory cytokines. This activates the coagulation cascade and promotes the formation of inflammatory and procoagulant particles by leukocytes, platelets, endothelial cells, and perivascular cells. Tissue factors activate the coagulation pathway through the production of factor VIIa, factor Xa, and thrombin (IIa). Thrombin facilitates the generation of C5a and C5b-9 when exposed to pathogenic agents and damaged host cells. These products continue to recruit inflammatory leukocytes, enhance the coagulation process, and release cytokines [3].

In sepsis, disseminated intravascular coagulation can range from subtle changes in laboratory tests to severe symptoms such as widespread clotting in blood vessels. The body's inflammatory response in sepsis triggers the coagulation system and fibrin deposition, which are important for fighting off infections. This inflammatory response also leads to activation of the coagulation system, inhibition of natural anticoagulant mechanisms, and suppression of fibrinolysis. Sepsis-induced disseminated intravascular coagulation disrupts circulation, leading to cellular activation and tissue damage. Bloodstream infections promote fibrin deposition through three main pathways: thrombin formation via tissue factor regulation, disruption of natural anticoagulant mechanisms, and impaired fibrin removal due to suppressed fibrinolysis [4].

Liver fibrosis is a common chronic condition caused by various factors. The liver is responsible for synthesizing most of the blood clotting factors, as well as anticoagulant substances and fibrinolytic enzymes. In liver fibrosis, liver function is impaired, leading to reduced production of these factors and complex coagulation disorders [5,6]. The liver synthesizes most of the blood clotting factors in the plasma, such as fibrinogen (factor I), factor V, factor XIII, and vitamin K-dependent factors like prothrombin, factors VII, IX, and X. Additionally, the liver also produces anticoagulant substances such as antithrombin III (ATIII), protein C, protein S, and some components of the fibrinolytic system, including plasminogen and α-antiplasmin. Patients with liver fibrosis generally have decreased overall protein synthesis. One of the consequences is a deficiency in vitamin K-dependent clotting factors II, VII, IX, and X [7–9]. Elevated vWF levels are mainly caused by increased release from endothelial cells and decreased serum activity of ADAMTS13, which is a type of metalloprotease enzyme that cleaves vWF. This enzyme is exclusively made in the liver, and its concentration in the serum decreases in patients with liver fibrosis [10]. The reduction of these factors can lead to various degrees of coagulation disorders.

The prothrombin time (PT) and activated partial thromboplastin time (aPTT) are two fundamental tests commonly utilized in clinical practice for screening and

diagnosing coagulation disorders. Prolonged aPTT results can be caused by various factors such as sepsis, liver dysfunction, deficiency of clotting factors, and the presence of lupus anticoagulants.

Various testing methods are available for fully automatic coagulation machines, including optical measurement techniques based on clot formation. When light is transmitted, it is absorbed by the clot. The optical measuring unit records the light transmission, and the photodetector captures the signal 10 times per second, converting it into a coagulation curve. This curve visually represents the reaction chain of coagulation factors in plasma. The normal coagulation curve typically has a sigmoid shape and consists of four parts: (1) Baseline stage: represents the activation of coagulation factors until thrombin is formed. During this stage, the optical density changes very little, resulting in a horizontal reaction curve; (2) Acceleration stage: corresponds to the process of thrombin converting fibrinogen (Fbg) to fibrin. Optical density changes rapidly during this stage, leading to a steep slope in the curve; (3) Deceleration stage: follows the acceleration stage and signifies that thrombin has almost fully converted Fbg into fibrin, causing a gradual decrease in optical density; (4) Endpoint stage: all Fbg is converted into fibrin, and the optical density stops changing. Algorithms used in the coagulation curve analysis include: (1) The first derivative, which represents the rate of fibrin clot formation. The peak of the first derivative corresponds to the largest clot formation rate and is used to calculate PT test results; (2) The second derivative, which represents the acceleration of fibrin clot formation. The peak of the second derivative also corresponds to the largest clot formation rate and is used to calculate aPTT, TT, endogenous factor, and dRVVT test results. The threshold algorithm determines the time when the optical density changes as Fbg is converted into fibrin.

There is increasing evidence that conventional coagulation tests such as PT and aPTT do not truly reflect the bleeding risk in patients with cirrhosis and sepsis [11]. These parameters are only descriptive in the first 5–10% of fibrin formation. However, the strength and stability of the clot as well as the interaction of all factors of the coagulation system with the vessel wall cannot be analyzed by these parameters [12,13]. Conventional coagulation tests make it difficult to predict the hemostasis of the patient and do not measure anticoagulant activity [14]. The coagulation curve can more comprehensively examine the coagulation process, from the beginning of clot formation to the end, which can provide more information about clot formation and fibrinolysis, increasing the ability of clinical diagnosis while observing what is happening in terms of testing and coagulation response. Therefore, this study was conducted to find the relationship between the following indices: aPTT (s), first derivative peak, second derivative peak, second derivative minimum peak, and delta-OD of the coagulation curve in patients diagnosed with cirrhosis, sepsis, and the control group.

## Materials and methods

### Study subject

This study aims to examine patients diagnosed with cirrhosis and sepsis who have normal aPTT test results. The patients are selected from the examination and treatment department, as well as the internal medicine department of Le Van Thinh Hospital from 8 January 2024–30 April 2024. All included patients are not receiving anticoagulant therapy at the time of testing, to avoid potential interference with aPTT and CWA results. Patients with incomplete information and those with prolonged aPTT test results are excluded from the study.

### Study design

A descriptive case series with a control group as follows:

1. Control group (Normal): This group consists of individuals visiting the examination department with health classification types 1 and 2 which according to national health screening guidelines, refers to individuals without acute or chronic illnesses, with normal clinical evaluations, and no laboratory abnormalities. All subjects in this group had normal aPTT values and no history of coagulopathy, ensuring that CWA results were minimally influenced by underlying health conditions.

2. Cirrhosis group: This group includes patients diagnosed with cirrhosis who have normal blood clotting results and visit the examination department for treatment. Patients in the cirrhosis group are diagnosed based on clinical signs, biochemical markers, and imaging findings consistent with the guidelines from the European Association for the Study of the Liver (EASL 2018). Diagnostic criteria included: evidence of chronic liver disease, ultrasound findings of liver surface nodularity or splenomegaly, and laboratory results showing elevated liver enzymes, hypoalbuminemia, or thrombocytopenia [15].

3. Sepsis group: This group comprises patients admitted to the internal medicine department for treatment with a sepsis diagnosis and normal blood clotting results. Patients in the sepsis group are diagnosed using the Third International Consensus Definitions for Sepsis and Septic Shock (Sepsis-3), as described by Singer et al (2016) [1]. Sepsis is defined as life-threatening organ dysfunction caused by a dysregulated host response to infection, quantified as an increase in the SOFA (Sequential Organ Failure Assessment) score of 2 points or more from baseline.

All subjects are tested for aPTT using the ACL Top 350CTS system (Werfen, USA), a fully automated coagulation analyzer equipped with a photo-optical clot-detection unit that monitors changes in optical absorbance at 405 nm or 671 nm to trace fibrin clot formation and generate derivative waveform curves. The assay used Hemosil APTT-SP reagent (Werfen, USA), which includes silica as the contact activator and exhibits high sensitivity to lupus anticoagulants.

## Variable definitions

- aPTT (s): Activated partial thromboplastin time

- aPTT (ratio): Ratio of patient aPTT value to the control aPTT value

- Max1: First derivative peak, indicating initial changes in thrombin production

- Max2: Second derivative peak, related to the second phase of the coagulation curve

- Min2: Second derivative minimum peak, related to the third phase of the coagulation curve

- Delta-OD: This measures the change in the optical density of a blood sample and reflects changes in the structure or properties of coagulation components.

These variables are represented in Fig 1 clot waveform analysis.

## Statistical analysis

In the hypothesis section of the study, the primary focus is on evaluating the aPTT and its influence on outcome measures. To achieve this objective, an analytical model comprising two tested models, namely the null model and the treatment model, is constructed and investigated.

For the Bayesian ANOVA analysis of three patient groups—control, cirrhosis, and sepsis—an equal prior probability of each model (0.5) is assumed. The Bayes Factor (BF10) can be interpreted as follows: the ratio of the likelihood of the data under the alternative hypothesis (H1, i.e., that there is a difference between groups—such as sepsis or cirrhosis) to the likelihood under the null hypothesis (H0, i.e., no difference, typically represented by the control group). A BF10 greater than 1 suggests increasing evidence for the alternative hypothesis over the null. BF10 from 3 to 10 suggests significant evidence in favor of the disease group (cirrhosis, sepsis), while BF10 from 10 to 30 indicates strong evidence. Moreover, BF10 from 30 to 100 implies very strong evidence, and BF10 > 100 denotes extreme evidence. A lower "Error %" value is preferable, particularly when it approaches 0, as it indicates a minimal degree of bias, reflecting a highly accurate evaluation method.

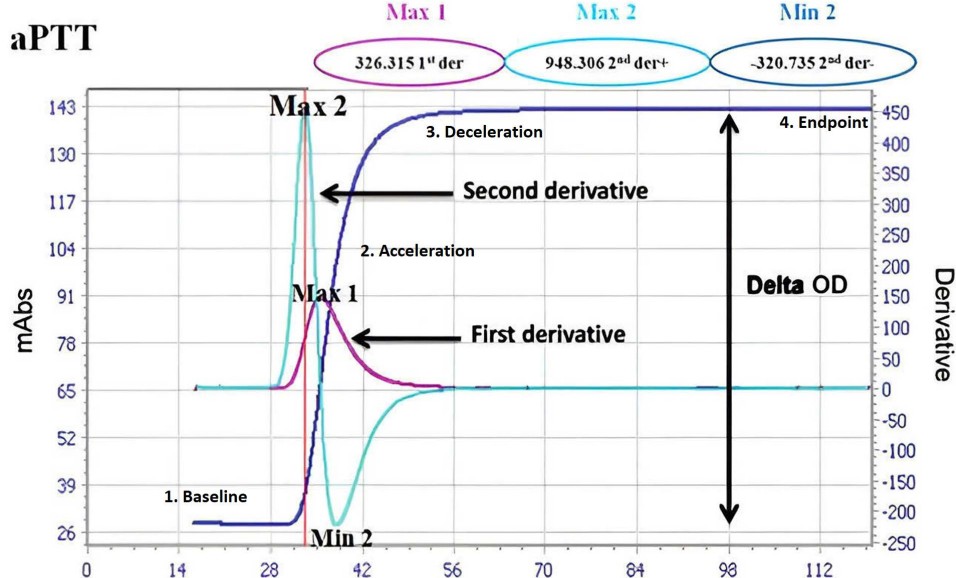

**Fig 1. Clot waveform analysis showing four distinct phases of clot formation.** 1. Baseline stage – initial flat region before clotting begins, indicating activation phase; 2. Acceleration stage – steep rise representing rapid conversion of Fbg to fibrin; 3. Deceleration stage – slowing slope as fibrin formation completes; 4. Endpoint stage – plateau where clot formation stabilizes.

The Effective Sample Size (ESS) serves as a metric for the number of truly independent samples in a Markov Chain Monte Carlo (MCMC). A higher ESS value signifies a larger number of independent samples, thereby enhancing the reliability of inferences drawn from MCMC.

The equal-tailed interval (ETI) and highest density interval (HDI) are utilized for establishing confidence intervals in probability distributions. To obtain stable estimates of HDI, the ESS should be at least 10 000. However, the ESS may be lower when ascertaining stable limits of ETI intervals, contingent upon the methods used for result measurement and reporting.

An analysis of the stability and influence of prior shape parameters on the BF can be conducted through a fluctuation analysis when the shape parameters vary from 0 to 2. Specifically, calculating the BF values for all shape parameters within the range of 0–2 facilitates the observation of fluctuations in the BF. In this analysis, the utilization of the Cauchy distribution enables the examination of scale changes and their consequential effects on the distribution and the BF.

In the comparative analysis between patient groups with cirrhosis or sepsis and the control group, the Bayesian method is employed to ascertain the BF value and assess the reliability of the hypothesis by comparing variable concentrations of aPTT (s), aPTT (ratio), Max1, Max2, Min2, and delta-OD.

### Ethical approval

This study was approved by the Ethics Committee of Le Van Thinh Hospital under Decision number: 01/HĐĐĐ-BVLVT. The need for informed consent was waived by the Ethics Committee due to the retrospective and fully anonymized nature of the data. All data used in this study were anonymized prior to analysis and publication. No identifying information (e.g., names, specific dates, group classifications, or ID numbers) was included in the shared data set. The anonymized data were used strictly for study purposes, and the authors had no access to identifiable participant information during or after the data analysis process.

## Results

### Characteristics of study sample

Table 1 presents the demographic and coagulation characteristics of the study population, including control, cirrhosis, and sepsis groups. The control group had a mean age of 39.6 years, significantly younger than the cirrhosis (63.5 years) and sepsis (71.5 years) groups. The gender distribution was relatively balanced across all groups. While the mean aPTT values were similar among the groups, variations were observed in CWA parameters. Notably, the Max1, Max2, and delta-OD values were markedly higher in the sepsis group compared to the control and cirrhosis groups, suggesting potential differences in coagulation dynamics despite normal aPTT results.

### Comparison of values between diseases

In order to compare values between different diseases, we utilized a Bayesian ANOVA model with MCMC sampling method. The model involved 2 chains, each with 10 000 iterations and a starting value of 5 000. We predicted variables aPTT (s), aPTT (ratio), Max1, Max2, Min2, and delta-OD across three disease groups: normal, cirrhosis, and sepsis. The model used the formula (variable~period – 1). This analysis helped in adjusting the posterior proportion for multiple testing by setting the prior probability of the initial hypothesis to 0.5 for all comparisons, following the approach outlined in the study by Westfall, Johnson, & Utts (1997).

**Table 1. Characteristics of study sample.**

|  | Control group (n = 35) | Sepsis group (n = 33) | Cirrhosis group (n = 27) |
|---|---|---|---|
| **Ages** |  |  |  |
| Mean (SD) | 39.6 (15.5) | 71.5 (15.5) | 63.5 (12.8) |
| Median [Min – Max] | 35.0 [19.0- 67.0] | 74.0 [27.0- 94.0] | 70.0 [36.0-79.0] |
| **Sex** |  |  |  |
| Male | 17 (48.6%) | 14 (42.4%) | 13 (48.1%) |
| Female | 18 (51.4%) | 19 (57.6%) | 14 (51.9%) |
| **aPTT (second)** |  |  |  |
| Mean (SD) | 31.1 (2.94) | 32.0 (2.91) | 31.4 (2.76) |
| Median [Min – Max] | 30.8 [26.0- 36.7] | 32.3 [26.4- 36.7] | 30.9 [27.4 - 36.2] |
| **aPTT (ratio)** |  |  |  |
| Mean (SD) | 1.02 (0.096) | 1.06 (0.098) | 1.03 (0.091) |
| Median [Min – Max] | 1.03 [0.85 - 1.20] | 1.06 [0.87 - 1.22] | 1.01 [0.89 - 1.18] |
| **Max1 (First derivative peak)** |  |  |  |
| Mean (SD) | 287 (64.5) | 394 (165) | 285 (135) |
| Median [Min – Max] | 282 [188- 545] | 382 [127 - 850] | 257 [112- 703] |
| **Max2 (Second derivative peak)** |  |  |  |
| Mean (SD) | 792 (151) | 1068 (435) | 833 (315) |
| Median [Min – Max] | 804 [487- 1110] | 1048 [294 - 2310] | 762 [359- 1635] |
| **Min2 (Second derivative minimum peak)** |  |  |  |
| Mean (SD) | 355 (80.7) | 335 (151) | 305 (136) |
| Median [Min – Max] | 348 [210 - 592] | 322 [89.5 - 809] | 291 [119- 695] |
| **Delta-OD (the change in the optical density)** |  |  |  |
| Mean (SD) | 300 (92.0) | 555 (307) | 317 (189) |
| Median [Min – Max] | 284 [145 - 652] | 472 [147 - 1583] | 275 [111 - 1029] |

Notes: SD: standard deviation, Min: minimum, Max: maximum.

## aPTT (s) parameter

After conducting post-analysis tests to compare different groups, the following results were obtained (Table 2).

When comparing the control group with the cirrhosis group: Prior odds for the difference between these two groups were 0.587. After observing the data, the posterior odds decreased to 0.165. The BF for the aPTT concentration in the cirrhosis group (BF10,U) was 0.281, indicating insignificant evidence strength. However, the difference was still found to be significant with an error level (error %) of 0.011.

Comparison of the control group with the sepsis group yielded the following results: The pretest probability ratio was 0.587. The posttest probability ratio decreased significantly to 0.309. The BF10 was 0.527, with an error % of 0.012, indicating a significant difference.

When comparing the cirrhosis group with the sepsis group: The pretest probability ratio was 0.587. The posttest probability ratio decreased to 0.211. The BF10 was 0.359, with an error % of 0.010, still showing a significant difference.

Fig 2 shows the density and distribution of aPTT in control, cirrhosis and sepsis groups using the raincloud plots.

**Table 2. Bayesian ANOVA test results for aPTT (s) value.**

| Variable | Patient group | Mean | Standard deviation | 95% Credible interval | |
|---|---|---|---|---|---|
| | | | | Lower | Upper |
| Intercept | | 31.486 | 0.299 | 30.896 | 32.073 |
| | Control | −0.341 | 0.371 | −1.082 | 0.379 |
| | Cirrhosis | −0.082 | 0.388 | −0.852 | 0.679 |
| | Sepsis | 0.423 | 0.377 | −0.309 | 1.179 |
| **Post Hoc Tests** | | | | | |
| | | Prior Odds | Posterior Odds | BF10,U | error % |
| Control | Cirrhosis | 0.587 | 0.165 | 0.281 | 0.011 |
| | Sepsis | 0.587 | 0.309 | 0.527 | 0.012 |
| Cirrhosis | Sepsis | 0.587 | 0.211 | 0.359 | 0.010 |

Notes: In Bayesian ANOVA analysis, this assumption states that the Posterior Probability and Prior Probability are equal. Prior Odds: This is the Prior Probability expressed as a ratio. Posterior Odds: This is the Posterior Probability expressed as a ratio. BF10,U: BF10 stands for "Bayes Factor", and "U" indicates that the evidence is unadjusted. Error %: The percentage of error, used to measure the precision of the analysis.

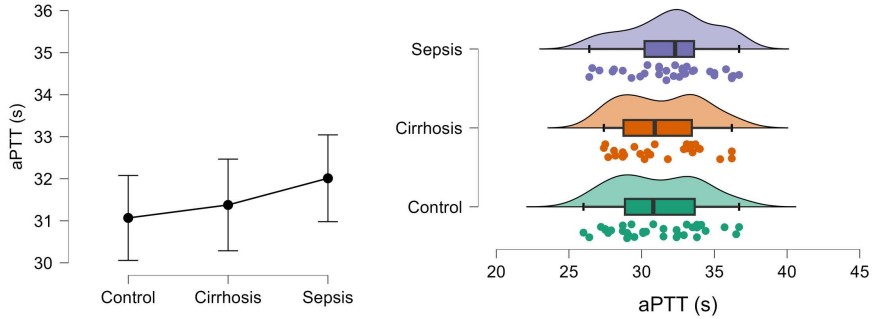

**Fig 2. Density and distribution of aPTT (s) in control, cirrhosis and sepsis groups.**

## aPTT parameter (ratio)

The control group with cirrhosis group: pretest probability ratio was 0.587, posttest probability ratio was 0.156, BF10 was 0.265, and error % was 0.011, still showing a statistically significant difference.

The control group with sepsis group: pretest probability ratio was 0.587, posttest probability ratio was 0.385, BF10 was 0.655, and error % was 0.011, this shows a statistically significant difference.

Cirrhosis group with sepsis group: pretest probability ratio was 0.587, posttest probability ratio was 0.296, BF10 was 0.504, and error level (error %) was 0.01 still showing a statistically significant difference (Table 3).

Fig 3 shows the density and distribution of aPTT (ratio) in control, cirrhosis and sepsis groups using the raincloud plots.

## Max1 parameter (1st derivative peak)

Control group vs. cirrhosis group: Prior Odds were 0.587, Posterior Odds were 0.153, BF10 was 0.261, and error % was 0.011, indicating a statistically significant difference.

Control group vs. sepsis group: Prior Odds were 0.587, Posterior Odds were 25.87, and BF10 was 44.042, showing an extreme difference between the two groups. The error % was small, only $8.301 \times 10^{-8}$, so the difference was considered highly significant.

Cirrhosis group vs. sepsis group: The pretest probability ratio was 0.587, and the post-test probability ratio increased to 3.432. The BF10 was 5.842, with an error % of $6.499 \times 10^{-7}$, indicating a significant difference (Table 4).

Fig 4 shows the density and distribution of Max1 in control, cirrhosis and sepsis groups using the raincloud plots.

**Table 3. Bayesian ANOVA test results for aPTT value (ratio).**

| Variable | Patient group | Mean | Standard deviation | 95% Credible interval | |
|---|---|---|---|---|---|
| | | | | Lower | Upper |
| Intercept | | 1.036 | 0.010 | 1.017 | 1.056 |
| | Control | −0.011 | 0.012 | −0.036 | 0.013 |
| | Cirrhosis | −0.007 | 0.013 | −0.033 | 0.019 |
| | Sepsis | 0.018 | 0.013 | −0.007 | 0.043 |
| **Post Hoc Tests** | | | | | |
| | | Prior Odds | Posterior Odds | BF10,U | error % |
| Control | Cirrhosis | 0.587 | 0.156 | 0.265 | 0.011 |
| | Sepsis | 0.587 | 0.385 | 0.655 | 0.011 |
| Cirrhosis | Sepsis | 0.587 | 0.296 | 0.504 | 0.010 |

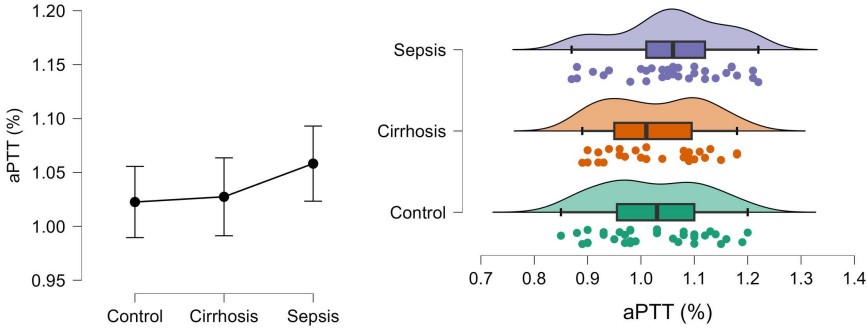

**Fig 3. Density and distribution of aPTT (ratio) in control, cirrhosis and sepsis groups.**

**Table 4. Bayesian ANOVA test results for Max1 value.**

| Variable | Patient group | Mean | Standard deviation | 95% Credible interval | |
|---|---|---|---|---|---|
| | | | | Lower | Upper |
| Intercept | | 322.110 | 13.316 | 295.872 | 348.238 |
| | Control | −31.670 | 17.336 | −65.971 | 2.125 |
| | Cirrhosis | −31.869 | 18.485 | −68.659 | 4.073 |
| | Sepsis | 63.539 | 18.503 | 27.342 | 100.107 |
| **Post Hoc Tests** | | | | | |
| | | Prior Odds | Posterior Odds | BF10,U | error % |
| Control | Cirrhosis | 0.587 | 0.153 | 0.261 | 0.011 |
| | Sepsis | 0.587 | 25.870 | 44.042 | 8.301 x 10−8 |
| Cirrhosis | Sepsis | 0.587 | 3.432 | 5.842 | 6.499 x 10−7 |

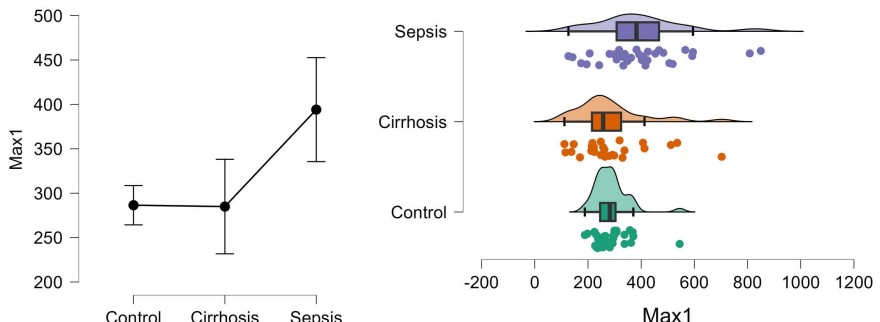

**Fig 4. Density and distribution of Max1 in control, cirrhosis and sepsis groups.**

## Max2 parameter (2nd derivative peak)

Table 5 shows the result of Max2 among groups as follows.

Control group with cirrhosis group: Prior Odds were 0.587, Posterior Odds were 0.185, BF10 was 0.316, and error % was 0.010, indicating a statistically significant difference.

**Table 5. Bayesian ANOVA test results for Max2 value.**

| Variable | Patient group | Mean | Standard deviation | 95% Credible interval | |
|---|---|---|---|---|---|
| | | | | Lower | Upper |
| Intercept | | 898.168 | 33.487 | 832.171 | 963.879 |
| | Control | −93.465 | 43.800 | −180.452 | −8.258 |
| | Cirrhosis | −55.241 | 45.884 | −146.552 | 34.200 |
| | Sepsis | 148.706 | 46.186 | 58.643 | 240.162 |
| **Post Hoc Tests** | | | | | |
| | | Prior Odds | Posterior Odds | BF10,U | error % |
| Control | Cirrhosis | 0.587 | 0.185 | 0.316 | 0.010 |
| | Sepsis | 0.587 | 22.780 | 38.782 | 9.631 x 10−8 |
| Cirrhosis | Sepsis | 0.587 | 1.478 | 2.517 | 0.009 |

The control group with sepsis group: Prior Odds were 0.587, Posterior Odds were 22.780, and BF10 was 38.782, indicating an extreme difference between the two groups. The error % was small, only 9.631 x 10−8, so the difference was considered very significant.

Cirrhosis group with sepsis group: pretest probability ratio was 0.587, the posttest probability ratio was 1.478, the BF10 was 2.517, and the error % was 0.009, showing that the difference was still significant.

Fig 5 shows the density and distribution of Max2 in control, cirrhosis and sepsis groups using the raincloud plots.

## Min2 parameter (2nd derivative minimum peak)

We compared the BF test for each group (Table 6).

Control group with cirrhosis group: Prior Odds were 0.587, Posterior Odds were 0.604, BF10 was 1.029, and error % was 0.009, indicating a statistically significant difference.

Control group with sepsis group: Prior Odds were 0.587, Posterior Odds were 0.180, BF10 was 0.306 indicating an extreme difference between the two groups. The error % was small, only 0.009, so the difference was considered very significant.

Cirrhosis group with sepsis group: pretest probability ratio was 0.587, the posttest probability ratio was 0.203, the BF10 was 0.345, and the error % was 0.010, showing that the difference was still significant.

Fig 6 shows the density and distribution of Min2 in control, cirrhosis and sepsis groups using the raincloud plots.

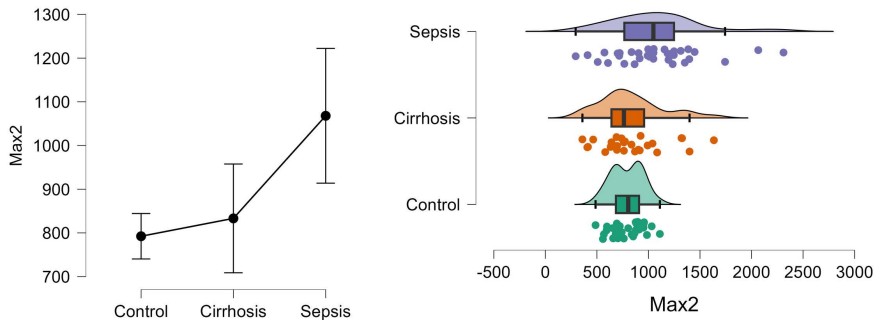

**Fig 5. Density and distribution of Max2 in control, cirrhosis and sepsis groups.**

**Table 6. Bayesian ANOVA test results for Min2 value.**

| Variable | Patient group | Mean | Standard deviation | 95% Credible interval | |
|---|---|---|---|---|---|
| | | | | Lower | Upper |
| Intercept | | 332.155 | 12.991 | 306.544 | 357.645 |
| | Control | 18.818 | 16.259 | −12.584 | 51.372 |
| | Cirrhosis | −21.140 | 17.234 | −55.822 | 12.047 |
| | Sepsis | 2.322 | 16.187 | −29.687 | 34.478 |
| **Post Hoc Tests** | | | | | |
| | | Prior Odds | Posterior Odds | BF10,U | error % |
| Control | Cirrhosis | 0.587 | 0.604 | 1.029 | 0.009 |
| | Sepsis | 0.587 | 0.180 | 0.306 | 0.013 |
| Cirrhosis | Sepsis | 0.587 | 0.203 | 0.345 | 0.010 |

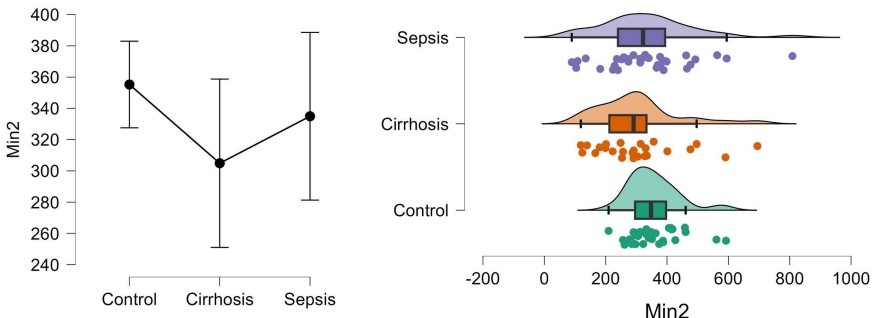

**Fig 6. Density and distribution of Min2 in control, cirrhosis and sepsis groups.**

## Delta-OD parameter

We compared by BF test for each group.

The control group with cirrhosis group: Prior Odds were 0.587, Posterior Odds were 0.167, BF10 was 0.284, and error % was 0.011, showing a statistically significant difference.

The control group with sepsis group: Prior Odds were 0.587, Posterior Odds were 735.298, and BF10 was 1251.781, showing an extreme difference between the two groups. The error % was small, only $3.701 \times 10^{-9}$, so the difference was considered significant.

Cirrhosis group with sepsis group: pretest probability ratio was 0.587, posttest probability ratio was 20.885, BF10 was 35.555, and error % was $1.668 \times 10^{-7}$, showing that the difference was still significant (Table 7).

Fig 7 shows the density and distribution of delta-OD in control, cirrhosis and sepsis groups using the raincloud plots.

## Discussion

This study investigated CWA parameters (including Max1, Max2, Min2, and delta-OD) in patients with cirrhosis and sepsis who presented with normal aPTT values. By focusing on cases without overt coagulopathy based on conventional aPTT, we aimed to evaluate the sensitivity of CWA in detecting subclinical changes in coagulation dynamics. The findings highlight important differences in CWA parameters across disease groups, supporting the potential of CWA as a supplementary diagnostic tool in clinical practice.

**Table 7. Bayesian ANOVA test results for delta-OD value.**

| Variables | Patient group | Mean | Standard deviation | 95% Credible interval | |
|---|---|---|---|---|---|
| | | | | Lower | Upper |
| Intercept | | 390.911 | 22.550 | 346.437 | 435.193 |
| | Control | −83.726 | 30.029 | −142.898 | −24.942 |
| | Cirrhosis | −66.861 | 31.851 | −129.889 | −4.550 |
| | Sepsis | 150.587 | 31.855 | 87.646 | 213.076 |
| **Post Hoc Tests** | | | | | |
| | | Prior Odds | Posterior Odds | BF10,U | error % |
| Control | Cirrhosis | 0.587 | 0.167 | 0.284 | 0.011 |
| | Sepsis | 0.587 | 735.298 | 1251.781 | $3.701 \times 10^{-9}$ |
| Cirrhosis | Sepsis | 0.587 | 20.885 | 35.555 | $1.668 \times 10^{-7}$ |

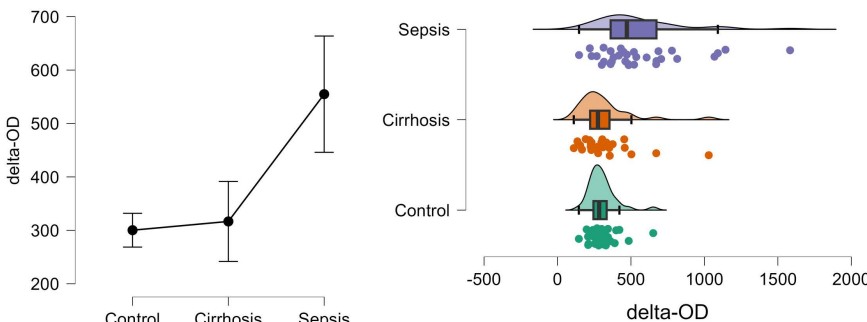

**Fig 7. Density and distribution of delta-OD in control, cirrhosis and sepsis groups.**

This study focused on cirrhosis and sepsis due to their high prevalence in our hospital setting and their well-documented associations with complex coagulation disturbances. In clinical practice at our center, PT and aPTT are commonly used as initial screening tools for coagulation abnormalities in these patient populations. However, many patients with cirrhosis or sepsis may present with normal aPTT values despite having underlying coagulopathies. To address this diagnostic gap, our study specifically targeted patients with normal aPTT results to evaluate the added value of CWA in detecting subclinical coagulation changes. This approach aims to support earlier recognition of coagulopathy and improve clinical decision-making in high-risk cases. While other conditions such as hemophilia or antiphospholipid syndrome also show distinct clot waveform patterns, their low incidence in our non-specialized general hospital setting limited their inclusion. Future studies conducted in hematology-focused centers may help broaden the comparative understanding of CWA across a wider spectrum of coagulation disorders.

Our study demonstrated significant differences in CWA parameters, specifically Max1, Max2, and delta-OD, among patients with cirrhosis, sepsis, and healthy controls, despite normal activated partial thromboplastin time (aPTT) values. Notably, the Max1, Max2, and delta-OD parameters were significantly elevated in patients with sepsis compared to both healthy controls and cirrhosis patients, indicating that CWA can detect subtle coagulation abnormalities that are not identified by traditional aPTT measurements. Specifically, delta-OD was markedly elevated in the sepsis group compared to both cirrhosis and control groups, with a BF10 of 1251.781, indicating extremely strong evidence of difference. These findings support previous reports highlighting the limited sensitivity of conventional coagulation tests such as PT and aPTT in capturing subtle and clinically relevant coagulation disturbances, particularly in critically ill patients or those with hepatic impairment [11,12,14].

Our findings are consistent with studies examining CWA in infectious conditions. Tan CW et al. (2021) reported significantly higher values of Min1 (6.48%/s vs. 5.05%/s, P < 0.001) and Min2 (0.92%/s² vs. 0.74%/s², P = 0.033) in patients with severe COVID-19 compared to mild cases, despite normal aPTT results, confirming the utility of CWA in early detection of coagulation changes during infection [16]. Tan CW et al. (2020) further demonstrated increased peak velocity and maximum acceleration in bacterial infections compared to controls (Min1: 6.36 vs. 5.74%/s, P = 0.024; Min2: 0.81 vs. 0.67%/s², P = 0.005), reinforcing the relevance of CWA in identifying hypercoagulability in infectious states [17]. Together, these findings highlight both studies demonstrate differences in coagulation status between patient groups, particularly when the disease is severe. Values like Min1, Min2, and Max2 all indicate increased coagulation in severely ill patients, consistent with the findings of our study regarding Max1 and Max2. However, the difference in delta-OD is only addressed in our study.

In the context of liver disease, Thanapirom et al. (2025) studied 560 cirrhotic patients and showed that several CWA parameters correlated with Child–Turcotte–Pugh (CTP) and End-Stage Liver Disease (MELD). In multivariable models, aPTT-based CWA indices were not independent predictors of 1-year mortality (Min1 aHR = 1.073; Min2 aHR = 2.122;

Max2 aHR = 1.511; all p > 0.05), whereas PT-based indices (lower Min2 and Max2) were independently predictive (Min2 aHR = 0.111, 95% CI 0.018–0.702; Max2 aHR = 0.062, 95% CI 0.007–0.553) [18]. These data support the broader utility of CWA in cirrhosis while suggesting that prognostic performance depends on the assay; our findings similarly show distinct group-level shifts in CWA parameters despite normal aPTT.

Regarding CWA parameters like Min1 (%/s), Min2 (%/s²), Max2 (%/s²), Tan CW's study showed that bacterial infections had higher CWA parameters compared to the control group, indicating a hypercoagulable state. Conversely, dengue infections had significantly lower CWA parameters, suggesting impaired global coagulation function [17]. In our study, Max1, Max2, and Min2 values were analyzed across different disease stages, revealing significant variations. Notably, Max1 was higher in patients with sepsis, Max2 varied significantly between the control, cirrhosis, and sepsis groups, and Min2 showed distinct differences, indicating specific coagulation changes.

The clinical implications of our results are substantial. Identifying early coagulation abnormalities using CWA parameters such as Max1, Max2, and delta-OD may enable clinicians to promptly intervene in septic or cirrhotic patients at risk of developing severe coagulation-related complications, thus potentially reducing morbidity and mortality. Given that conventional assays such as aPTT often remain within normal ranges despite underlying coagulopathy, integration of CWA into routine clinical practice could significantly enhance diagnostic accuracy and therapeutic decision-making.

Although our study utilized robust Bayesian statistical methods and clearly defined patient populations, larger, multi-center studies would further confirm and generalize these findings. Subsequent research could focus on establishing relationships between these CWA parameters and clinical outcomes, such as bleeding events or thrombotic complications, to validate their prognostic capabilities comprehensively.

This study has several limitations. First, PT and Fbg levels were not included, despite their importance in interpreting aPTT and CWA results. In our clinical setting, PT and Fbg tests are not routinely performed in all patients, and Fbg assays are not reimbursed under the national health insurance system, which limits their accessibility. Although previous studies have reported correlations between delta-OD and Fbg concentration [19], and Fbg is known to affect APTT and its CWA parameters [20], we were unable to evaluate this relationship due to the absence of Fbg data. Second, other coagulation and fibrinolytic markers such as D-dimer and thrombin-antithrombin complex (TAT) were not assessed, primarily due to their high cost and limited availability in routine hospital testing. Third, the relatively small sample size reduced the statistical power for advanced analyses such as receiver operating characteristic (ROC) curves, which could have provided further insight into the diagnostic performance of CWA parameters. Lastly, the control group consisted of younger, healthier individuals attending routine health screenings, whereas the cirrhosis and sepsis groups were older, reflecting the demographics of these conditions. Given previous evidence that age and gender may weakly influence CWA parameters [21], this difference in age distribution represents a potential confounding factor. Future studies with larger, age-matched cohorts and expanded laboratory parameters—including PT, Fbg, D-dimer, and TAT—will be designed to overcome these limitations and provide a more comprehensive and accurate evaluation of CWA in clinical settings.

## Conclusion

The Max1, Max2, and delta-OD parameters can effectively reflect the status of coagulation disorders in patients with liver fibrosis and septicemia, even when the aPTT test remains within normal limits. These results open up the potential application of coagulation curve indices in the diagnosis and monitoring of coagulation disorders in these patient groups. This helps clinical physicians choose appropriate treatment protocols and preventive measures for coagulation-related complications in each patient group.

## Supporting information

**S1 File. Contains values supporting tables and figures.**
(PDF)

## Acknowledgments

We acknowledge Nguyen Tat Thanh University, Ho Chi Minh City, Vietnam for supporting this study. We thank all the members of the medical laboratory of Le Van Thinh hospital for helping us.

## Author contributions

**Conceptualization:** Tran Qui Phuong Linh, Le Minh Thuan.

**Formal analysis:** Le Minh Thuan.

**Investigation:** Tran Qui Phuong Linh, Nguyen Thi Nhan.

**Methodology:** Nguyen Thi Nhan, Le Minh Thuan, Le Trung Phuong.

**Supervision:** Tran Qui Phuong Linh.

**Validation:** Nguyen Thi Nhan, Le Trung Phuong.

**Writing – original draft:** Tran Qui Phuong Linh, Nguyen Thi Nhan.

**Writing – review & editing:** Le Minh Thuan, Le Trung Phuong.

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
