## [Decision Letter · Decision Letter 0]

31 Jan 2025

PONE-D-24-41352SURVEY OF CLOT WAVEFORM ANALYSIS OF NORMAL ACTIVATED PARTIAL THROMBOPLASTIN TIME IN PATIENTS WITH CIRRHOSIS AND SEPSIS AT LE VAN THINH HOSPITALPLOS ONE

Dear Dr. Nguyen,

Thank you for submitting your manuscript to PLOS ONE. After careful consideration, we feel that it has merit but does not fully meet PLOS ONE’s publication criteria as it currently stands. Therefore, we invite you to submit a revised version of the manuscript that addresses the points raised during the review process. Subject: Decision on Manuscript PONE-D-24-41352 – Minor Revision

Dear Nhan Thi Nguyen,

We have received the reports from our advisors on your manuscript titled Survey of Clot Waveform Analysis of Normal Activated Partial Thromboplastin Time in Patients with Cirrhosis and Sepsis at Le Van Thinh Hospital (Manuscript Number: PONE-D-24-41352), which you submitted to PLOS ONE.

Based on the advice received, the Editors feel that your manuscript could be reconsidered for publication, provided that minor revisions are incorporated.

When preparing your revised manuscript, please carefully address the reviewers’ comments, which are attached. Additionally, you are requested to submit a detailed response to each of the reviewers' comments. Please check online for any attached reviewer files.

Submission Guidelines for the Revised Manuscript:

Upload two identical versions of the revised manuscript:

One version should include all revisions highlighted in colored text for easy identification.

The other should be a clean version without highlights.

Submit your editable source files (e.g., Word, TeX).

Upload the response to the reviewers as a separate submission item under ‘Attachment to Manuscript.’

Please ensure that all required modifications are incorporated before resubmission.

Best regards,

Dr. Kovuri Umadevi

Please submit your revised manuscript by Mar 17 2025 11:59PM. If you will need more time than this to complete your revisions, please reply to this message or contact the journal office at plosone@plos.org . Please include the following items when submitting your revised manuscript:

We look forward to receiving your revised manuscript.

Kind regards,

Kovuri Umadevi

Academic Editor

PLOS ONE

Journal Requirements:

3. We note you have included a table to which you do not refer in the text of your manuscript. Please ensure that you refer to Tables 1-7 in your text; if accepted, production will need this reference to link the reader to the Table.

Reviewers' comments:

Reviewer's Responses to Questions

**Comments to the Author**

1. Is the manuscript technically sound, and do the data support the conclusions?

Reviewer #1: Yes

2. Has the statistical analysis been performed appropriately and rigorously? 

Reviewer #1: Yes

3. Have the authors made all data underlying the findings in their manuscript fully available?

Reviewer #1: Yes

4. Is the manuscript presented in an intelligible fashion and written in standard English?

Reviewer #1: Yes

5. Review Comments to the Author

Reviewer #1: I have read your article with great attention. i find it very authentic. a few points to bear in mind:

1. Line 80: The three Full stops (...) at the end of the sentence are not necessary.

2. Image resolution needs to be improved.

6. PLOS authors have the option to publish the peer review history of their article (what does this mean? ). If published, this will include your full peer review and any attached files.

**Do you want your identity to be public for this peer review?** For information about this choice, including consent withdrawal, please see our Privacy Policy .

Reviewer #1: No

---

## [Author Response · Author response to Decision Letter 1]

16 Mar 2025

Rebuttal letter

For the manuscript “Survey of clot waveform analysis of normal activated partial thromboplastin time in patients with cirrhosis and sepsis at Le Van Thinh hospital” [PONE-D-24-41352]

Response to reviewers

We sincerely appreciate the valuable time and effort the reviewers and editors have dedicated to evaluating our manuscript. Their insightful comments and constructive feedback have contributed significantly to enhancing the clarity, rigor, and overall quality of our work. We have carefully addressed all concerns and have made substantial revisions to align with the expectations for major revisions.

To ensure transparency and clarity, we have provided a detailed, point-by-point response to each comment. Each response is structured and corresponds directly to the respective reviewer’s query. Additionally, we have made significant improvements in the writing throughout the manuscript to enhance readability and better communicate the core aspects and contributions of our study.

1. Response to Academic Editor:

Authors’ response: That’s included

Authors’ response: That’s included

Authors’ response: That’s included

2. Response to Journal requirements:

Comment-1: Please ensure that your manuscript meets PLOS ONE's style requirements, including those for file naming.

Authors’ response: Thank you very much for your comments. We formatted the manuscript based on the instruction guidelines defined in the template.

Comment-2: We note that your Data Availability Statement is currently as follows: [All relevant data are within the manuscript and its Supporting Information files.]

Please confirm at this time whether or not your submission contains all raw data required to replicate the results of your study. Authors must share the “minimal data set” for their submission. PLOS defines the minimal data set to consist of the data required to replicate all study findings reported in the article, as well as related metadata and methods

Authors’ response: Thank you for bringing this up. We confirm that the minimal dataset required to replicate the results of our study has been uploaded as part of the Supporting Information files accompanying the manuscript. This dataset includes:

- A table of values used to build means, standard deviations, other measures reported and graphs.

Comment-3: We note you have included a table to which you do not refer in the text of your manuscript. Please ensure that you refer to Tables 1-7 in your text; if accepted, production will need this reference to link the reader to the Table.

Authors’ response: We apologize for not mentioning all the tables in the manuscript's text. We have added and referenced them in the manuscript.

Comment-4: Please review your reference list to ensure that it is complete and correct. If you have cited papers that have been retracted, please include the rationale for doing so in the manuscript text, or remove these references and replace them with relevant current references. Any changes to the reference list should be mentioned in the rebuttal letter that accompanies your revised manuscript. If you need to cite a retracted article, indicate the article’s retracted status in the References list and also include a citation and full reference for the retraction notice

Authors’ response: We reviewed the reference list. After that, we replaced the reference “National Guideline. National Institute for Health and Care Excellence: Guidelines. Cirrhosis in Over 16s: Assessment and Management. London: National Institute for Health and Care Excellence (NICE)” (reference No.5 in old manuscript) by “Bitto N, Liguori E, La Mura V. Coagulation, Microenvironment and Liver Fibrosis. Cells. 2018;7(8):85.” (reference No.5) and “Airola C, Pallozzi M, Cerrito L, Santopaolo F, Stella L, Gasbarrini A, et al. Microvascular Thrombosis and Liver Fibrosis Progression: Mechanisms and Clinical Applications. Cells. 2023;12(13):1712.” (reference No.6). Besides, we have added one reference “Kujovich JL. Coagulopathy in liver disease: a balancing act. Hematology American Society of Hematology Education Program. 2015;2015:243-9.” (reference No.9). We have carefully double-checked the reference list and confirm its completeness and accuracy. We verified that none of the cited papers have been retracted.

2. Response to Reviewer-1:

Comment-1: Line 80: The three Full stops (...) at the end of the sentence are not necessary.

Authors’ response: We have removed the unnecessary ellipsis at the end of the sentence in line 80.

Comment-2: Image resolution needs to be improved

Authors’ response: We have enhanced the resolution of all figures to meet the publication standards of PLOS ONE by using the PLOS-provided tool PACE. High-quality images have been uploaded in compliance with the journal’s guidelines.

---

## [Decision Letter · Decision Letter 1]

29 Apr 2025

PONE-D-24-41352R1Survey of clot waveform analysis of normal activated partial thromboplastin time in patients with cirrhosis and sepsis at Le Van Thinh hospitalPLOS ONE

Dear Dr. Nhan,

Thank you for submitting your manuscript to PLOS ONE. After careful consideration, we feel that it has merit but does not fully meet PLOS ONE’s publication criteria as it currently stands. Therefore, we invite you to submit a revised version of the manuscript that addresses the points raised during the review process.

We look forward to receiving your revised manuscript.

Kind regards,

Kovuri Umadevi

Academic Editor

PLOS ONE

Journal Requirements:

Additional Editor Comments:

Subject: Decision on Manuscript PONE-D-24-41352R1 – Major Revision

Dear Dr. Nhan,

We have completed the review process for your manuscript titled "Survey of clot waveform analysis of normal activated partial thromboplastin time in patients with cirrhosis and sepsis at Le Van Thinh Hospital" (Manuscript Number: PONE-D-24-41352R1).

Based on the reviewers' comments and evaluations, the editorial decision is Major Revision.

We kindly request you to carefully address all the reviewers' comments and submit a revised version of your manuscript.

Thank you for submitting your work to PLOS ONE. We look forward to receiving your revised manuscript.

Best regards,

Dr. Kovuri Umadevi

Academic Editor

PLOS ONE

Reviewers' comments:

Reviewer's Responses to Questions

**Comments to the Author**

1. If the authors have adequately addressed your comments raised in a previous round of review and you feel that this manuscript is now acceptable for publication, you may indicate that here to bypass the “Comments to the Author” section, enter your conflict of interest statement in the “Confidential to Editor” section, and submit your "Accept" recommendation.

Reviewer #2: (No Response)

Reviewer #3: (No Response)

2. Is the manuscript technically sound, and do the data support the conclusions?

Reviewer #2: Partly

Reviewer #3: Yes

3. Has the statistical analysis been performed appropriately and rigorously? 

Reviewer #2: Yes

Reviewer #3: Yes

4. Have the authors made all data underlying the findings in their manuscript fully available?

Reviewer #2: Yes

Reviewer #3: Yes

5. Is the manuscript presented in an intelligible fashion and written in standard English?

Reviewer #2: Yes

Reviewer #3: Yes

6. Review Comments to the Author

Reviewer #2: The authors investigated the relationship between the clot waveform analysis parameters in APTT and some disorders. The data is interesting and valuable. However, there are some points to revise in the manuscript.

- The authors included control, cirrhosis and sepsis groups in the study. Please describe the detailed diagnosis criteria in cirrhosis and sepsis groups. Because there are some guidelines for these groups, and the authors should cite the guidelines and define the inclusion criteria in the selected patients.

- Why did the authors include only patients with normal APTT clotting times? The data of the comparison between normal APTT and abnormal APTT in each disorder would be helpful to find the tendencies in clot waveform analysis. Please consider adding this comparison data.

- Although there are some explanations about cirrhosis and sepsis, please describe the reason why the authors selected these two groups as the target in this study. As they mentioned, other disorders such as hemophilia, antiphospholipid syndrome etc. shows APTT prolongation, and it was reported that the clot waveforms in these disorders were different from those of normal (Matsumoto et al. Haemophilia 2017, Matsumoto et al. Int J Hematol 2016).

- In the introduction, the following sentence was described.

“Prolonged aPTT results can be caused by various factors such as sepsis, liver dysfunction, deficiency of clotting factors, and systemic lupus erythematosus.” I think “lupus anticoagulants” word is more suitable instead of systemic lupus erythematosus.

- The authors divided into four parts including baseline, acceleration, deceleration and endpoint stages and explained in the introduction section. It is better to add this explanation in Figure 1 for readers’ understanding.

- aPTT(%) parameter is shown. I think this is not percentage but ratio. Please consider the rewording.

- Do the authors have PT and Fbg data? Please add these data in Table 1. PT and Fbg results are important to interpret APTT data.

- It was reported that delta-OD correlated Fbg concentration (Suzuki et al. Thromb Res 2019). Delta-OD can also indicate Fbg concentration in this study. Please show the Fbg concentration and confirm the correlation between delta-OD and Fbg concentrations.

- The authors analyzed data based on Post Hoc Tests. However, ROC analysis is also required to find the usefulness in the differentiation among three groups.

- The comparison between clot waveform analysis parameters and other markers like D-dimer, TAT is also meaningful to show the usefulness of clot waveform analysis. Even if APTT is not prolonged, other markers may show higher level than normal range.

Reviewer #3: The paper and research present important clinical evidence to the use of clot waveform analysis (CWA) and potentially advancing CWA towards clinical care. A few comments and suggested revisions before publishing:

Research design

1. Noted that the samples included patients from control, cirrhosis and sepsis groups. Suggest to elaborate on the health classification types 1 and 2 for normal samples briefly to provide clarity that APTT and the subsequent CWA are minimally affected by the health status.

2.Fibrinogen is known to affect APTT and its CWA (Ref: Siegemund, T., Scholz, U., Schobess, R. & Siegemund, A. Clot waveform analysis in patients with haemophilia A. Hamostaseologie 34(Suppl 1), S48-52 (2014)). CWA levels that are adjusted to the fibrinogen levels of the subjects are now available in other platforms. Suggest to justify that fibrinogen levels do not affect the reported results; either through validating that the fibrinogen levels are not significantly different, or to make a note if the reported CWA in your platform have already taken into account the fibrinogen differences.

3. Suggest to also make a note on the anticoagulation status of the disease populations, just also to ensure that the CWA collected are not influenced by any drug use.

4. Age and gender have also been reported to be weakly correlated to CWA (Ref: Wong, W.H.; Tan, C.W.; Abdul Khalid, N.B.; Dalimoenthe, N.Z.; Yip, C.; Tantanate, C.; Lim, R.D.; Kim, J.H.; Ng, H.J. Reagent Effects on the Activated Partial Thromboplastin Time Clot Waveform Analysis: A Multi-Centre Study. Diagnostics 2023, 13, 2447.). Noted that the demographics of the disease groups are significantly different from the control groups, hence suggest to perform age-adjusted analysis.

Writing

1. Line 84: Suggest to change "S-shape" to "sigmoid shape" as other platforms reporting CWA might also present the data in a sigmoid but in the opposite direction depending on whether it is optical absorbance or transmittance.

2. Materials and methods: Please re-write this section in present tense (e.g "The patients are selected...", "This group consists of...") instead of the future tense.

3. Materials and methods: Consider to give one or two lines of introduction to the analysis method (ACL TOP350CTS) focusing on its optical technology and how it relates to the clot formation, as well as the specifications of the reagent used (Hemosil APTT-SP) such as the activator content, lupus anticoagulant sensitivity etc. This will allow the potential reader to understand how CWA from other platforms might be similar or dissimilar in interpretation in the same patient populations.

4. Line 144: Consider to rewrite the definition of BF10. Does it mean that the B10 is a comparison of the disease group vs the control group?

5. Line 169: Please ensure that the Decision No. does not contain typo error.

6.Line 205: Post test probability ratio had "decreased" from 0.587 to 0.211.

7. Discussion: suggest to quote the literature papers as "Tan CW et al" instead of full name.

8. Line 304: Suggest not to use "In conclusion..." as there is further discussion after this paragraph. You might consider consolidating this paragraph with the Conclusion section.

9. Discussion: Suggest to focus the discussion on the current CWA findings and reference back to the literature, to draw more on the relevance of this study, as well as how the current findings would advance our understanding or use of CWA in clinical settings.

In general, the findings presented by the authors are interesting, novel and very informative for clinical care when CWA is poised to value-add routine coagulation results. These findings will allow clinicians to understand how to interpret CWA in different patient populations and important to adoption of the technology.

7. PLOS authors have the option to publish the peer review history of their article (what does this mean? ). If published, this will include your full peer review and any attached files.

**Do you want your identity to be public for this peer review?** For information about this choice, including consent withdrawal, please see our Privacy Policy .

Reviewer #2: No

Reviewer #3: No

---

## [Author Response · Author response to Decision Letter 2]

11 Jul 2025

Rebuttal letter

For the manuscript “Survey of clot waveform analysis of normal activated partial thromboplastin time in patients with cirrhosis and sepsis at Le Van Thinh hospital” [PONE-D-24-41352]

Response to reviewers

We sincerely appreciate the valuable time and effort the reviewers and editors have dedicated to evaluating our manuscript. Their insightful comments and constructive feedback have contributed significantly to enhancing the clarity, rigor, and overall quality of our work. We have carefully addressed all concerns and have made substantial revisions to align with the expectations for major revisions.

To ensure transparency and clarity, we have provided a detailed, point-by-point response to each comment. Each response is structured and corresponds directly to the respective reviewer’s query. Additionally, we have made significant improvements in the writing throughout the manuscript to enhance readability and better communicate the core aspects and contributions of our study.

1. Response to Reviewer #2:

Comment-1: The authors included control, cirrhosis and sepsis groups in the study. Please describe the detailed diagnosis criteria in cirrhosis and sepsis groups. Because there are some guidelines for these groups, and the authors should cite the guidelines and define the inclusion criteria in the selected patients.

Authors’ response:

Thank you for your valuable comment. We have now included detailed diagnostic criteria for cirrhosis and sepsis in the "Materials and Methods" section of the manuscript. For cirrhosis, we referred to the EASL Clinical Practice Guidelines (2018), and for sepsis, we used the Sepsis-3 definitions as proposed by Singer et al. (2016). These references have also been added to the reference list.

Comment-2: Why did the authors include only patients with normal APTT clotting times? The data of the comparison between normal APTT and abnormal APTT in each disorder would be helpful to find the tendencies in clot waveform analysis. Please consider adding this comparison data

Authors’ response:

Thank you for your insightful comment. In clinical practice at our hospital, patients with cirrhosis and sepsis often present with coagulopathy, which can be life-threatening if undetected. However, traditional coagulation tests such as PT and aPTT are often the primary methods used to assess coagulation status. These tests may not detect early or subtle coagulation changes. Therefore, the focus of our study was to investigate whether clot waveform analysis (CWA) could reveal coagulation abnormalities even in patients who show normal aPTT values. By selecting patients with normal aPTT, we aimed to assess the sensitivity of CWA in detecting early-stage or hidden coagulopathy in cirrhosis and sepsis populations. This approach underscores the potential clinical value of CWA as a supplementary diagnostic tool beyond conventional assays.

Comment-3: Although there are some explanations about cirrhosis and sepsis, please describe the reason why the authors selected these two groups as the target in this study. As they mentioned, other disorders such as hemophilia, antiphospholipid syndrome etc. shows APTT prolongation, and it was reported that the clot waveforms in these disorders were different from those of normal (Matsumoto et al. Haemophilia 2017, Matsumoto et al. Int J Hematol 2016).

Authors’ response:

Thank you for this important suggestion. Our hospital, Le Van Thinh General Hospital, is a frontline multi-specialty institution without a hematology specialty unit. Therefore, patients with rare hematologic disorders such as hemophilia or antiphospholipid syndrome are rarely encountered and typically referred to specialized centers. In contrast, we routinely treat patients with sepsis and cirrhosis, both of which are frequently associated with coagulopathy. These two conditions also reflect distinct pathophysiological mechanisms of coagulation dysfunction — inflammatory-induced changes in sepsis and synthetic failure in cirrhosis — making them valuable models for evaluating the sensitivity of clot waveform analysis (CWA) even in the presence of normal aPTT. This practical and clinical relevance guided our selection of these two patient groups.

Comment-4: In the introduction, the following sentence was described.

“Prolonged aPTT results can be caused by various factors such as sepsis, liver dysfunction, deficiency of clotting factors, and systemic lupus erythematosus.” I think “lupus anticoagulants” word is more suitable instead of systemic lupus erythematosus.

Authors’ response:

Thank you for pointing this out. We agree with your suggestion and have revised the sentence in the Introduction accordingly. The term “lupus anticoagulants” is indeed more accurate in this context, as it directly refers to the antibodies that cause prolonged aPTT, rather than the broader clinical diagnosis of systemic lupus erythematosus. The revised sentence now reads:

“Prolonged aPTT results can be caused by various factors such as sepsis, liver dysfunction, deficiency of clotting factors, and the presence of lupus anticoagulants.”

Comment-5: The authors divided into four parts including baseline, acceleration, deceleration and endpoint stages and explained in the introduction section. It is better to add this explanation in Figure 1 for readers’ understanding.

Authors’ response:

We have revised Figure 1 to include visual annotations that clearly indicate the four stages of clot formation: baseline, acceleration, deceleration, and endpoint. These labels have been added directly to the figure to enhance readers' understanding of the clot waveform phases as described in the Introduction. The updated figure has been included in the revised manuscript.

Comment-6: aPTT(%) parameter is shown. I think this is not percentage but ratio. Please consider the rewording.

Authors’ response:

Thank you for your observation. We agree with your assessment and have revised the terminology from “aPTT (%)” to “aPTT ratio” throughout the manuscript. We also updated the definition in the “Variable definitions” section to clarify that this parameter represents the ratio of patient aPTT value to control aPTT value (aPTT_patient / aPTT_control), rather than a percentage.

Comment-7: Do the authors have PT and Fbg data? Please add these data in Table 1. PT and Fbg results are important to interpret APTT data.

Authors’ response:

Thank you for this valuable suggestion. Unfortunately, neither PT nor Fbg results were available for the participants in our study. At our institution, PT and Fbg testing are not routinely performed for all patients with cirrhosis or sepsis, especially in cases where aPTT results appear normal. Moreover, Fbg testing is not covered by the national health insurance scheme in Vietnam, which further limits its use in routine clinical practice. As a result, these data were not collected and are not included in Table 1. We acknowledge this as a limitation of our study and have added a corresponding statement in the Discussion section. Future studies incorporating both PT and Fbg measurements will be important to provide a more comprehensive assessment of coagulation abnormalities and their relationship with CWA parameters.

Comment-8: It was reported that delta-OD correlated Fbg concentration (Suzuki et al. Thromb Res 2019). Delta-OD can also indicate Fbg concentration in this study. Please show the Fbg concentration and confirm the correlation between delta-OD and Fbg concentrations.

Authors’ response:

Thank you for pointing out this important relationship. We are aware of the reported correlation between delta-OD and fibrinogen concentration, as described by Suzuki et al. (2019). However, fibrinogen levels were not available in our study dataset, and we were therefore unable to assess this correlation directly. We have acknowledged this limitation in the Discussion and emphasized the potential value of including fibrinogen measurements in future research to further explore the diagnostic utility of delta-OD in clot waveform analysis.

Comment-9: The authors analyzed data based on Post Hoc Tests. However, ROC analysis is also required to find the usefulness in the differentiation among three groups.

Authors’ response:

Thank you for your insightful recommendation. We agree that ROC analysis can provide valuable information in evaluating the diagnostic utility of specific parameters. However, due to the relatively small sample size in our study, we were concerned that ROC curves may not yield statistically robust or generalizable results. We have therefore limited our analysis to Bayesian post hoc comparisons, which are more appropriate given the current dataset. Nevertheless, we acknowledge the potential value of ROC analysis and recommend it as a direction for future studies with larger sample sizes to validate the discriminatory power of CWA among different patient groups.

Comment-10: The comparison between clot waveform analysis parameters and other markers like D-dimer, TAT is also meaningful to show the usefulness of clot waveform analysis. Even if APTT is not prolonged, other markers may show higher level than normal range.

Authors’ response:

Thank you for your helpful suggestion. We agree that comparing CWA parameters with markers such as D-dimer and TAT could provide additional insight into the clinical utility of CWA. However, in our current study, these markers were not included due to practical limitations. At our hospital, the high cost of D-dimer and TAT assays limits their routine use, especially in settings where resources are constrained. Consequently, we did not have access to sufficient data to perform these comparisons. We acknowledge this as a limitation in the Discussion section and agree that future studies incorporating these biomarkers would further enhance the evaluation of CWA in detecting subclinical coagulopathies.

2. Response to Reviewer #3:

Comment-1: Noted that the samples included patients from control, cirrhosis and sepsis groups. Suggest to elaborate on the health classification types 1 and 2 for normal samples briefly to provide clarity that APTT and the subsequent CWA are minimally affected by the health status.

Authors’ response:

We have revised the description of the control group in the Research Design section to clarify that “health classification types 1 and 2” refer to individuals without known medical conditions or laboratory abnormalities, according to national health screening guidelines. We also specified that all control subjects had normal aPTT values and no history of coagulopathy, ensuring minimal interference with CWA results.

Comment-2: Fibrinogen is known to affect APTT and its CWA (Ref: Siegemund, T., Scholz, U., Schobess, R. & Siegemund, A. Clot waveform analysis in patients with haemophilia A. Hamostaseologie 34(Suppl 1), S48-52 (2014)). CWA levels that are adjusted to the fibrinogen levels of the subjects are now available in other platforms. Suggest to justify that fibrinogen levels do not affect the reported results; either through validating that the fibrinogen levels are not significantly different, or to make a note if the reported CWA in your platform have already taken into account the fibrinogen differences.

Authors’ response:

We are aware that fibrinogen levels can influence clot waveform analysis (CWA) parameters, particularly delta-OD, as reported by Siegemund et al. (2014). However, similar to the limitation noted in our responses to Reviewer #2 (comment-7 and 8), fibrinogen testing was not performed in our study population. This is due to the fact that fibrinogen assays are not routinely ordered in our hospital for patients with cirrhosis or sepsis—particularly when aPTT results are within the normal range—and such testing is not covered under the national health insurance scheme in Vietnam. We have acknowledged the absence of fibrinogen data as a limitation in the Discussion section. Future research should include fibrinogen measurements to assess their impact on CWA parameters and improve interpretive accuracy.

Comment-3: Suggest to also make a note on the anticoagulation status of the disease populations, just also to ensure that the CWA collected are not influenced by any drug use.

Authors’ response:

Thank you for your observation. We have clarified in the “Materials and Methods” section that all enrolled patients were not receiving anticoagulant therapy at the time of testing. This exclusion was implemented to minimize confounding effects and ensure that clot waveform analysis (CWA) results reflected the patients’ intrinsic coagulation status.

Comment-4: Age and gender have also been reported to be weakly correlated to CWA (Ref: Wong, W.H.; Tan, C.W.; Abdul Khalid, N.B.; Dalimoenthe, N.Z.; Yip, C.; Tantanate, C.; Lim, R.D.; Kim, J.H.; Ng, H.J. Reagent Effects on the Activated Partial Thromboplastin Time Clot Waveform Analysis: A Multi-Centre Study. Diagnostics 2023, 13, 2447.). Noted that the demographics of the disease groups are significantly different from the control groups, hence suggest to perform age-adjusted analysis.

Authors’ response:

Thank you for this valuable comment. We acknowledge that age and gender may influence clot waveform analysis (CWA) parameters, as noted by Wong et al. (2023). In our study, the control group was composed of generally healthy individuals attending health screenings, and therefore tended to be younger than patients in the cirrhosis or sepsis groups. Due to the nature of the hospital setting, it was not feasible to recruit healthy controls matched in age to the disease groups. While this introduces a potential confounding factor, we have noted it as a limitation in the Discussion. Future studies with larger and age-matched populations are needed to validate and extend our findings.

3. Writing

Comment-1: Line 84: Suggest to change "S-shape" to "sigmoid shape" as other platforms reporting CWA might also present the data in a sigmoid but in the opposite direction depending on whether it is optical absorbance or transmittance.

Authors’ response:

Thank you for your helpful suggestion. We agree that “sigmoid shape” is a more precise and technically appropriate term than “S-shape” in this context. We have revised the sentence accordingly to improve clarity and consistency with other CWA platforms.

Comment-2: Materials and methods: Please re-write this section in present tense (e.g "The patients are selected...", "This group consists of...") instead of the future tense.

Authors’ response:

We have revised the entire “Materials and Methods” section to use the present tense, in accordance with standard scientific writing conventions. All instances of future tense have been replaced to ensure consistency and clarity in describing the study procedures.

Comment-3: Materials and methods: Consider to give one or two lines of introduction to the analysis method (ACL TOP350CTS) focusing on its optical technology and how it relates to the clot formation, as well as the specifications of the reagent used (Hemosil APTT-SP) such as the activator content, lupus anticoagulant sensitivity etc. This will allow the potential reader to understand how CWA from other platforms might be similar or dissimilar in interpretation in the same patient populations.

Authors’ response:

Thank you for your suggestion. We have added a short description in the Materials and Methods section to introduce the ACL TOP 350CTS system and the Hemosil APTT-SP reagent. This addition aims to provide context for interpreting CWA results and comparing them with findings from other analytical platforms.

Comment-4: Line 144: Consider to rewrite the definition of BF10. Does it mean that the B10 is a comparison of the disease group vs the control group?

Authors’ response:

Thank you for your suggestion. We agree that the previous definition of BF10 lacked clarity. We have revised the sentence to clarify that BF10 represents the ratio of the likelihood of the data under the alternative hypothesis (e.g., difference in a disease group) to the likelihood under the null hypothesis (e.g., control group). This change aims to ensure clearer understanding of the Bayesian interpretation.

Comment-5: Line 169: Please ensure that the Decisio

---

## [Decision Letter · Decision Letter 2]

7 Aug 2025

PONE-D-24-41352R2Survey of clot waveform analysis of normal activated partial thromboplastin time in patients with cirrhosis and sepsis at Le Van Thinh hospitalPLOS ONE

Dear Dr. Nhan,

Thank you for submitting your manuscript to PLOS ONE. After careful consideration, we feel that it has merit but does not fully meet PLOS ONE’s publication criteria as it currently stands. Therefore, we invite you to submit a revised version of the manuscript that addresses the points raised during the review process.

We look forward to receiving your revised manuscript.

Kind regards,

Kovuri Umadevi

Academic Editor

PLOS ONE

Journal Requirements:

**Additional Editor Comments:**

Dear Nguyen Thi Nhan,

We are writing to inform you that we have received the required reviewer feedback for your revised manuscript titled:

"Survey of clot waveform analysis of normal activated partial thromboplastin time in patients with cirrhosis and sepsis at Le Van Thinh hospital"

(Manuscript ID: PONE-D-24-41352R2) submitted to PLOS ONE.

After evaluation of the reviewer comments and editorial assessment, the decision on your manuscript is: Minor Revision.

We request you to address the remaining minor concerns raised by the reviewers and submit a revised version of your manuscript along with a detailed point-by-point response.

We look forward to your resubmission.

Best regards,

Dr. Kovuri Umadevi

Academic Editor

PLOS ONE

Reviewers' comments:

Reviewer's Responses to Questions

**Comments to the Author**

1. If the authors have adequately addressed your comments raised in a previous round of review and you feel that this manuscript is now acceptable for publication, you may indicate that here to bypass the “Comments to the Author” section, enter your conflict of interest statement in the “Confidential to Editor” section, and submit your "Accept" recommendation.

Reviewer #2: All comments have been addressed

Reviewer #3: All comments have been addressed

2. Is the manuscript technically sound, and do the data support the conclusions?

Reviewer #2: Yes

Reviewer #3: Yes

3. Has the statistical analysis been performed appropriately and rigorously? 

Reviewer #2: Yes

Reviewer #3: Yes

4. Have the authors made all data underlying the findings in their manuscript fully available?

Reviewer #2: Yes

Reviewer #3: Yes

5. Is the manuscript presented in an intelligible fashion and written in standard English?

Reviewer #2: Yes

Reviewer #3: Yes

6. Review Comments to the Author

Reviewer #2: (No Response)

Reviewer #3: Thank you for taking the feedback seriously and making significant improvements to the clarity of the publication.

Just a few minor language suggestions to consider.

1. Suggest to remove the ":" from these few lines since the referred tables or figures might not follow the line (subject to the editor's discretion) - Lines 220, 269, 283, 297.

2. Minor typo error on Line 382: "which limis their accessibility" to "which limits their accessibility".

7. PLOS authors have the option to publish the peer review history of their article (what does this mean? ). If published, this will include your full peer review and any attached files.

**Do you want your identity to be public for this peer review?** For information about this choice, including consent withdrawal, please see our Privacy Policy .

Reviewer #2: No

Reviewer #3: No

---

## [Author Response · Author response to Decision Letter 3]

17 Aug 2025

Rebuttal letter

For the manuscript “Survey of clot waveform analysis of normal activated partial thromboplastin time in patients with cirrhosis and sepsis at Le Van Thinh hospital” [PONE-D-24-41352R2]

Response to reviewers

We sincerely appreciate the valuable time and effort the reviewers and editors have dedicated to evaluating our manuscript. Their insightful comments and constructive feedback have contributed significantly to enhancing the clarity, rigor, and overall quality of our work. We have carefully addressed all concerns and have made substantial revisions to align with the expectations for minor revisions.

To ensure transparency and clarity, we have provided a detailed, point-by-point response to each comment. Each response is structured and corresponds directly to the respective reviewer’s query. Additionally, we have made significant improvements in the writing throughout the manuscript to enhance readability and better communicate the core aspects and contributions of our study.

1. Response to Journal Requirements:

Comment-1: If the reviewer comments include a recommendation to cite specific previously published works, please review and evaluate these publications to determine whether they are relevant and should be cited. There is no requirement to cite these works unless the editor has indicated otherwise.

Authors’ response:

In line with the journal’s guidance, we reviewed the works suggested by the reviewers and cited those that are directly relevant and improve the interpretation of our findings. Specifically, we added:

- [19] Suzuki A, Suzuki N, Kanematsu T, Shinohara S, Arai N, Kikuchi R, et al. Clot waveform analysis in Clauss fibrinogen assay contributes to classification of fibrinogen disorders. Thrombosis Research. 2019;174:98-103: Discussion, paragraph 9.

- [20] Siegemund T, Scholz U, Schobess R, Siegemund A. Clot waveform analysis in patients with haemophilia A. Hamostaseologie. 2014;34 Suppl 1:S48-52: Discussion, paragraph 9.

- [21] Wong WH, Tan CW, Abdul Khalid NB, Dalimoenthe NZ, Yip C, Tantanate C, et al. Reagent Effects on the Activated Partial Thromboplastin Time Clot Waveform Analysis: A Multi-Centre Study. Diagnostics (Basel, Switzerland). 2023;13(14) : Discussion, paragraph 9.

Comment-2: Please review your reference list to ensure that it is complete and correct. If you have cited papers that have been retracted, please include the rationale for doing so in the manuscript text, or remove these references and replace them with relevant current references. Any changes to the reference list should be mentioned in the rebuttal letter that accompanies your revised manuscript. If you need to cite a retracted article, indicate the article’s retracted status in the References list and also include a citation and full reference for the retraction notice

Authors’ response:

Thank you for your insightful comment. In accordance with the journal’s guidance, we reviewed the entire reference list for completeness and accuracy and screened all citations for retraction status as of August 17, 2025. We confirm that no cited items have been retracted.

We added several current and directly relevant publications to strengthen context and interpretation, as listed below and reflected in the revised manuscript and clean reference list.

- [15]. Angeli P, Bernardi M, Villanueva C, Francoz C, Mookerjee RP, Trebicka J, et al. EASL Clinical Practice Guidelines for the management of patients with decompensated cirrhosis. Journal of Hepatology. 2018;69(2):406-60 – added to Study design, paragraph 2.

- [18]. Thanapirom K, Suksawatamnuay S, Thaimai P, Ananchuensook P, Kijrattanakul P, Angchaisuksiri P, et al. Association between Clot Waveform Analysis Parameters and the Severity of Liver Cirrhosis. Thrombosis and haemostasis. 2025 – added to Discussion, paragraph 5.

- [19]. Suzuki A, Suzuki N, Kanematsu T, Shinohara S, Arai N, Kikuchi R, et al. Clot waveform analysis in Clauss fibrinogen assay contributes to classification of fibrinogen disorders. Thrombosis Research. 2019;174:98-103 – added to Discussion, paragraph 9.

- [20]. Siegemund T, Scholz U, Schobess R, Siegemund A. Clot waveform analysis in patients with haemophilia A. Hamostaseologie. 2014;34 Suppl 1:S48-52 – added to Discussion, paragraph 9.

- [21]. Wong WH, Tan CW, Abdul Khalid NB, Dalimoenthe NZ, Yip C, Tantanate C, et al. Reagent Effects on the Activated Partial Thromboplastin Time Clot Waveform Analysis: A Multi-Centre Study. Diagnostics (Basel, Switzerland). 2023;13(14) – added to Discussion, paragraph 9.

2. Response to Reviewer #3:

Comment-1: Suggest to remove the ":" from these few lines since the referred tables or figures might not follow the line (subject to the editor's discretion) - Lines 220, 269, 283, 297.

Authors’ response:

We agree with the reviewer. We have removed the colon following at lines 220, 269, 283, and 297. The changes are shown in the tracked-changes version.

Comment-2: Minor typo error on Line 382: "which limis their accessibility" to "which limits their accessibility".

Authors’ response:

We agree with the reviewer. We have corrected the typo on Line 382 from “which limis their accessibility” to “which limits their accessibility.” We reviewed the manuscript for similar minor errors and corrected them; no further instances remain in the revised version.

---

## [Decision Letter · Decision Letter 3]

6 Oct 2025

Survey of clot waveform analysis of normal activated partial thromboplastin time in patients with cirrhosis and sepsis at Le Van Thinh hospital

PONE-D-24-41352R3

Dear Dr. Nguyen Thi Nhan

We’re pleased to inform you that your manuscript has been judged scientifically suitable for publication and will be formally accepted for publication once it meets all outstanding technical requirements.

Kind regards,

Kovuri Umadevi

Academic Editor

PLOS ONE

Additional Editor Comments (optional):

Dear Nguyen Thi Nhan,

We have now received the required number of reviewer reports for your manuscript entitled:

“Survey of Clot Waveform Analysis of Normal Activated Partial Thromboplastin Time in Patients with Cirrhosis and Sepsis at Le Van Thinh Hospital” (Manuscript Number: PONE-D-24-41352R3), which you submitted to PLOS ONE.

Based on the reviewers’ assessments and the Editorial Board’s evaluation, I am pleased to inform you that the decision is to accept your manuscript for publication.

Your study makes a valuable contribution to the field, and we are delighted to see it published in PLOS ONE. On behalf of the Editorial Board, I thank you for choosing our journal to disseminate your work.

Congratulations on this achievement.

Sincerely,

Dr. Kovuri Umadevi

Academic Editor

PLOS ONE

Reviewers' comments:

Reviewer's Responses to Questions

**Comments to the Author**

1. If the authors have adequately addressed your comments raised in a previous round of review and you feel that this manuscript is now acceptable for publication, you may indicate that here to bypass the “Comments to the Author” section, enter your conflict of interest statement in the “Confidential to Editor” section, and submit your "Accept" recommendation.

Reviewer #2: All comments have been addressed

Reviewer #3: (No Response)

2. Is the manuscript technically sound, and do the data support the conclusions?

Reviewer #2: Yes

Reviewer #3: Yes

3. Has the statistical analysis been performed appropriately and rigorously? 

Reviewer #2: Yes

Reviewer #3: Yes

4. Have the authors made all data underlying the findings in their manuscript fully available?

Reviewer #2: Yes

Reviewer #3: Yes

5. Is the manuscript presented in an intelligible fashion and written in standard English?

Reviewer #2: Yes

Reviewer #3: Yes

6. Review Comments to the Author

Reviewer #2: (No Response)

Reviewer #3: Thank you for addressing my concerns and my previous suggested edits. The manuscript is now recommended for publication.

7. PLOS authors have the option to publish the peer review history of their article (what does this mean? ). If published, this will include your full peer review and any attached files.

**Do you want your identity to be public for this peer review?** For information about this choice, including consent withdrawal, please see our Privacy Policy .

Reviewer #2: No

Reviewer #3: No

---

## [Editor Report · Acceptance letter]

PONE-D-24-41352R3

PLOS ONE

Dear Dr. Nhan,

I'm pleased to inform you that your manuscript has been deemed suitable for publication in PLOS ONE. Congratulations! Your manuscript is now being handed over to our production team.

Kind regards,

on behalf of

Dr. Kovuri Umadevi

Academic Editor

PLOS ONE